# *Arctium lappa* Extract Suppresses Inflammation and Inhibits Melanoma Progression

**DOI:** 10.3390/medicines6030081

**Published:** 2019-07-29

**Authors:** Bruno A. C. Nascimento, Luiz G. Gardinassi, Inaê M. G. Silveira, Marília G. Gallucci, Mariana A. Tomé, Júlia Fernanda D. Oliveira, Mirella R. A. Moreira, Alyne F. G. Meirelles, Lúcia H. Faccioli, Cristiane Tefé-Silva, Karina F. Zoccal

**Affiliations:** 1Centro Universitário Barão de Mauá (CBM), Rua Ramos de Azevedo, n 423, 14090-180 Ribeirão Preto, SP, Brazil; 2Departamento de Análises Clínicas, Toxicológicas e Bromatológicas. Faculdade de Ciências Farmacêuticas de Ribeirão Preto, Universidade de São Paulo (FCFRP-USP), Avenida do Café, s/n, 14040-903 Ribeirão Preto, São Paulo, Brazil

**Keywords:** *Arctium lappa*, natural product, phytomedicine, inflammation, melanoma

## Abstract

**Background:** Arctium lappa has been used as popular medicinal herb and health supplement in Chinese societies. Bioactive components from *A. lappa* have attracted the attention of researchers due to their promising therapeutic effects. In this study, we investigated the effects of *A. lappa* hydroalcoholic extract (Alhe) during different models of inflammation, in vivo. **Methods:** The anti-inflammatory activity was evaluated through the air pouch model. For this, mice received an inflammatory stimulus with lipopolysaccharide (LPS) and were later injected with Alhe. To assess anti-tumoral activity, the animals were inoculated with B16F10 cells and injected with Alhe every 5 days, along the course of 30 days. Controls were submitted to the same conditions and injected with the vehicle. Peritoneal or air pouch fluids were collected to evaluate leukocyte counting or cellular activation via quantification of cytokines and nitric oxide. **Results:** Alhe injection reduced the neutrophil influx and production of inflammatory mediators in inflammatory foci after LPS or tumor challenges. Furthermore, Alhe injection reduced tumor growth and enhanced mice survival. **Conclusions:** Collectively, these data suggest that Alhe regulates immune cell migration and activation, which correlates with favorable outcome in mouse models of acute inflammation and melanoma progression.

## 1. Introduction

Inflammation is a complex process operating in response to noxious stimuli and tissue damage. Acute inflammation is characterized by tissue infiltration and accumulation of leukocytes, such as neutrophils and monocytes, and increasing vascular permeability that promotes protein extravasation and edema formation [1,2]. The observed alterations occur due to the production and release of pro-inflammatory (e.g., TNF-α, IL-1β, IL-6, prostaglandins) and anti-inflammatory (e.g., IL-10, TGF-β, leukotrienes) mediators [1]. However, excessive or unbalanced inflammation might lead to tissue damage and become counterproductive. 

Cancer is a major cause of death worldwide, while the inflammatory process is tightly connected to cancer progression [3]. Among several cancer types, skin cancer is characterized by an imbalance towards reduced apoptosis or increased cell proliferation and survival in the epidermis [4,5], combined with a failure of the immune system to contain the tumoral development. Although UV radiation is the leading cause of skin cancer, other causative agents include viruses, mutagens in food, mutagens in chemicals and genetic susceptibility [6,7]. Currently, skin cancer is treated by surgical removal, radiation therapy, chemotherapy, or cryosurgery [4]. Cancer chemotherapy faces many problems, such as severe adverse effects and multi-drug resistance formation [8,9,10,11]. In line with disadvantages in conventional chemotherapy, research on complementary and alternative medicine offers great potential for innovation in cancer therapy [12,13]. Phytochemical compounds extracted from plant roots, bulbs, barks, leaves, and stems have shown significant anti-tumoral activity and are promising candidates for innovate chemotherapy [4]. Medicinal plants and their natural constituents have long been used worldwide in folk medicine, to treat inflammatory processes of diverse origins. Particularly in Brazil, endemic plant species or their extracts are widely used by traditional communities, for the same goal [14,15,16]. 

*Arctium lappa* L. (popularly known as burdock) is widely used in popular medicine worldwide as a diuretic and antipyretic agent and has also been used to treat hypertension, gout, hepatitis, and other inflammatory disorders [17,18]. Interest in *A. lappa* extracts has grown due to their ample therapeutic potential [19,20,21]. *A. lappa* extracts contain several compounds, including flavonoids, lignans, tannins, phenolic acids, alkaloids, and terpenoids [20]. Interestingly, lignans from *A. lappa* exhibit antiproliferative and apoptotic effects over leukemic cells [22], and also have anti-tumoral effects on pancreatic cancer cell lines [23]. Moreover, the main active compound of *A. lappa* L., Arctigenin, have shown to exhibit anti-inflammatory effect in a mouse model of colitis, operating via the inhibition of MAPK and NF-κB pathways [24].

In this study, we investigated the effects of Alhe injection on different models of inflammation, in vivo. We observed a significant reduction of neutrophil infiltration and production of inflammatory mediators after LPS or tumoral cell injection in mice treated with Alhe. Furthermore, Alhe injection had a striking impact on melanoma progression and mice mortality. 

## 2. Materials and Methods

### 2.1. Hydroalchoolic Arctium lappa L. Extract

The extract of *Arctium lappa* was obtained from 100 g of ground and dry bark, dissolved in 1 L of 70% ethanol (JT Baker, Belo Horizonte, Brazil). A dark bottle conditioned at room temperature was used for 72 h, with daily shaking, and the extract was then filtered on filter paper. The filtrate was dried using a rotary evaporator (Eppendorf Vacuum Concentrator Plus, Hamburg, Germany). The dried extract was weighed and suspended in 997 μL PBS (phosphate-buffered saline) and 3 μL of DMSO (Dimethyl sulfoxide, Sigma-Aldrich, Saint Louis, MO, USA), until a final concentration of 1 g/mL was obtained. This extract was filtered and stored in 500 μL aliquots at −20 °C in a freezer. Plant samples were collected in the city of Uberaba, state of Minas Gerais, Brazil (Lat. −19.743623, Long −47.828514). The specimens were analyzed and identified by Prof. Dr. Milton Groppo and deposited in the herbarium of the Faculdade de Filosofia, Ciências e Letras of Ribeirão Preto, Universidade de São Paulo.

### 2.2. Animals

Male or female Balb/c mice (6–8 weeks old) were maintained at the animal facility of Centro Universitário Barão de Mauá (São Paulo, Brazil). Mice were maintained at 25 °C, with a 12 h/12 h light/dark cycle, and were provided with free access to food and water. All experiments were approved and conducted in accordance with the guidelines of the Ethics Committee on Research and Animal Experimentation of Centro Universitário Barão de Mauá (process #280/17; 13 April 2017).

### 2.3. Determination of Alhe Dose and Leukocyte Count in Blood and the Peritoneal Cavity

Different groups of mice (n = 5) were injected with 300 μL of *Alhe* at different concentrations (50, 300, and 500 mg/kg) via the intraperitoneal route (i.p.). The animals received the extract every 5 days, in a total of 30 days, for chronic effect. The controls mice group received 300 μL of sterile PBS + ethanol via the i.p. route, in the same ratio of dilution of the extract. Following this, the animals were euthanatized in a CO_2_ chamber and the blood was immediately collected. Lavage of the peritoneal cavity was performed by injecting 3 mL of sterile PBS into the peritoneum, which was gently massaged for 1 min. Peritoneal fluid was collected using a syringe with a needle inserted into the inguinal region. Peripheral blood cells and total peritoneal cells were counted with Turk’s solution, using Neubauer chambers. Differential leukocyte counts were performed on cytospin preparations stained with a commercial kit based on the Romanowsky staining procedure (Panótico_ Laborclin, Paraná, Brazil). The experiments were performed twice.

### 2.4. Air Pouch Induction and Stimuli Injection

Prior to creating the air pouch, animals were anesthetized (i.p.) with 100 μL of a ketamine (80 mg/kg) and xylasine (15 mg/kg) solution. The dorsal region of the mice were shaved and 3 mL of sterile air was injected subcutaneously with a 25 G needle, through a sterile filter (Millex^®^ GV 0.22 μm—Merck Millipore Ltd, Tullagreen, IRL). Three days later, 2 mL of sterile air was injected into the pre-existing air pouch, as previously described [25]. On the 6th day, after the first air injection, the animals received 1 mL of LPS (1 μg/mL) or PBS into the air pouch. After 1 h, 1 mL of PBS or Alhe (300 mg/kg/pouch) were injected. Air pouches injected with sterile PBS were used as negative controls. After 4 h of injection, the animals were euthanized and 2 mL of sterile PBS containing 3.6% sodium citrate were injected into the air pouches, which were gently massaged for 1 min. Lavage fluid was collected to count cells and quantify inflammatory mediators. The experiments were performed twice.

### 2.5. Anti-Tumoral Assay

The animals (n = 5/experimental group) had their dorsal region trichotomized. On the following day, cultured B16F10 murine melanoma cells exhibiting an exponential growth phase were adjusted to the concentration of 10^6^ cells/mL in 0.9% saline solution. Tumoral cells were implanted subcutaneously with 10^6^ cells/0.1 mL of 0.9% saline. After 4 h of the tumor inoculation, the mice received PBS or Alhe (50 mg/kg by v.i.p.) every 5 days for 30 days. Mice were weighed before tumor inoculation and every 5 days after tumor challenge. Survival was also monitored throughout the 30 days. The remaining animals were euthanized to collect blood and the peritoneal fluid. The experiments were performed twice.

### 2.6. Quantification of Nitric Oxide (NO) and Total Protein

Nitrite (NO_2_^–^) was quantified in cell-free fluid (peritoneal or air pouch cavities) as an indicator of NO production by the Griess method [26]. The amount of nitrite in the samples was obtained by a standard curve using serial NaNO_2_ dilutions. The absorbance at 540 nm was recorded 10 min after addition of NaNO_2_. Total protein extravasation was quantified in cell-free fluid (peritoneal or air pouches cavities) with Coomassie protein assay reagent (Pierce Chemical, Rockford, AZ, USA), according to the manufacturer’s instructions.

### 2.7. Cytokine Quantification

The cell-free fluid (peritoneal or air pouches cavities) obtained was used to quantify IL-1β, TNF-α, and IL-6 with commercial ELISA kits (R&D Systems, Minneapolis, MN, USA), according to the manufacturer´s instructions. Optical densities were measured at 405 nm in a microplate reader (µQuant^™^, Biotek^®^ Instruments, Inc., Winooski, VT, USA).

### 2.8. Tumor Evaluation

Tumors were measured with a Vernier caliper, and their size (in mm^3^) was calculated, as previously described [27]. Briefly, tumor volume = (length × height × width)/2. For this experiment, mice were monitored for 30 days after tumor injection.

### 2.9. Statistical Analyses

Statistical analyses were performed using the GraphPad software v 5.0 (GraphPad, San Diego, CA, USA). Data were expressed as mean ± standard error of mean (SEM). The differences between any two groups were evaluated using two-tailed Student’s *t*-test. For comparison of multiple groups, we performed one-way analysis of variance (ANOVA), followed by Tukey’s post-hoc test. Differences in survival rates were analyzed using the log-rank test. Values of *p* < 0.05 were considered significant. 

## 3. Results

### 3.1. Dose-Dependent Effects of Alhe on Local and Peripheral Leukocyte Accumulation

To determine the influences of Alhe during physiological condition, Balb/c mice were injected with different Alhe doses (50, 300, and 500 mg/kg) via the i.p. route, every 5 days for 30 days. Individual weights were comparable before injection, while Alhe administration at any dose did not impact weight gain over time (data not shown). Alhe injection at doses of 300 and 500 mg/kg induced a significant influx of total leukocytes and neutrophils into the peritoneal cavity (Figure 1A,B, respectively). We also observed increased leukocytosis in the peripheral blood, at doses 300 and 500 mg/Kg (Figure 1C), but neutrophil counts increased only at dose 500 mg/Kg (Figure 1D). Based on these data, Alhe at 300 mg/kg was used to evaluate its biological activity in different models of inflammation.

### 3.2. Alhe Inhibits Leukocyte Recruitment and Activation in Response to LPS Challenge

Although different *A. lappa* extracts and isolated compounds exhibit anti-inflammatory properties, the mechanisms accounting for these effects are unknown. To investigate whether leukocyte infiltration into inflammatory foci is affected, we used an air pouch model of LPS-induced inflammation. Compared to the control animals, total leukocyte counts increased substantially after 4 h of LPS injection into air pouches (Figure 2A). Further cell stratification demonstrated that neutrophil was the major leukocyte subset accumulating into the air pouches (Figure 2B). Mononuclear cell counts were similar to the control group (Figure 2C). Notably, Alhe reduced the total leukocyte and neutrophil influx into air pouches, whereas mononuclear cell counts remained unaltered (Figure 2A–C).

The significant impact of Alhe over local neutrophilic accumulation provides a first evidence of its anti-inflammatory activity to LPS challenge. High levels of total proteins accumulated in inflammatory sites as a consequence of the increased vascular permeability, which reflected the edema formation. LPS challenge induced significant increase in levels of total protein, while Alhe suppressed edema formation (Figure 2D). Importantly, Alhe inhibited the production of IL-6, TNF-α, and IL-1β in the LPS-stimulated animals (Figure 2E–G, respectively). These cytokines play major roles in the different processes, such as neutrophil recruitment and edema formation. 

### 3.3. Alhe Restricts Melanoma Progression and Mortality

Inflammation is a critical feature of cancer, which not only drives the oncogenic transformation of epithelial cells, but promotes growth, progression, and metastasis. Therefore, we sought to determine the effects of *Alhe* injection in a model of skin cancer. For this purpose, Balb/c mice were subcutaneously implanted with 1 × 10^6^ B16F10 cells (melanoma). These animals were injected with Alhe (50 mg/kg) or PBS, for 30 days. Tumor implantation induced significant accumulation of total leukocytes and neutrophils in the peritoneal cavity, which was suppressed by Alhe injection (Figure 3A,B). Furthermore, Alhe injection had no effect on total leukocyte accumulation on peripheral blood (Figure 3C), but it reduced the number of circulating neutrophils after tumor implantation (Figure 3D). We also observed that melanoma implantation promoted edema in the peritoneal cavity (Figure 3E) and induced the production of several inflammatory mediators (Figure 3F–I). Of note, Alhe injection reduced edema formation (Figure 3E) and suppressed the release of NO (Figure 3F), IL-6 (Figure 3G), TNF-α (Figure 3H), and IL-1β (Figure 3I) in the peritoneal cavity. Together, these results indicated that Alhe injection exerts significant effects on the inflammatory processes, suggesting a potential anti-tumoral activity. To test this hypothesis, mice were subcutaneously implanted with 1 × 10^6^ B16F10 cells and subsequently injected with *Alhe* or PBS for 30 days. Strikingly, *Alhe* injection suppressed the tumor growth by 38% after 20 days (Figure 4A) and enhanced mice survival after 30 days (Figure 4B). 

## 4. Discussion

The search for new compounds with anti-inflammatory and anti-tumoral activity with minimal side effects is challenging. Many unexplored natural sources with significant biological activity have spurred research to fill this gap. In this study, we confirmed that *A. lappa* exhibits robust anti-inflammatory properties in different modalities of inflammation. Importantly, we provide a proof of concept for its protective activity against melanoma progression and mortality, *in vivo*. However, future studies are necessary to identify specific compounds and the associated biological activities, to understand their mechanisms of action. It will be important to delineate which class of molecules or whether the mixture of different compounds is necessary for its therapeutic effects. Moreover, detailed evaluation of whether the effects of Alhe injection observed herein correlate with the effects of oral administration of Alhe is also necessary. This is particularly important because intraperitoneal injections would not be appropriate for treatment of patients, whereas oral treatment would be the preferred route for *A. lappa* administration in humans.

The significant impact of Alhe injection after LPS-challenge, an acute model of inflammation, indicates a rapid effect that is able to regulate neutrophil recruitment, cytokine production, and edema formation. These processes are critical for the control and elimination of pathogens, but deregulated responses might cause tissue damage and exacerbate disease [28,29]. TNF-α, IL-6, and IL-1β are pleiotropic cytokines that regulate a broad range of biological events, including cell differentiation, proliferation, tissue development, and death, as well as innate and adaptive immune responses [30,31]. All three cytokines contribute to excessive tissue accumulation of neutrophils during inflammation. During rheumatoid arthritis, neutrophils accumulate and destroy the synovial tissue in response to TNF-α and IL-6 [32,33,34]. Moreover, we have shown that IL-1β is necessary for neutrophil accumulation in the lungs, edema formation, and death, after scorpion envenomation [35,36]. Data reported herein also support other findings showing the beneficial effect of *A. lappa* extract during diverse inflammatory diseases, such as arthritis, lung inflammation, cardiovascular diseases, asthma, and cancer [19,37,38].

*A. lappa* extract has been shown to reduce cell viability, causing apoptosis and also reducing proliferation of the breast cancer cell lines MCF7 and MDA-MB-231 [39]. Another study found anti-proliferative activity of *A. lappa* extract against additional cell lines, such as K562 (leukemia) and 786-0 (renal cancer) [40]. In addition, *A. lappa* root extract exhibits protective properties against toxic substances, lowering cell mutations. Most studies related to *A. lappa* anti-tumoral activity were performed with Arctigenin and Arctiin, lignans that influences intracellular pathways targeted for anti-tumoral therapies. Arctigenin is able to eliminate nutrient-deprived cancer cells [22], to inhibit liver cancer tumorigenesis [41], and to repress melanin synthesis in B16BL6 melanoma cells [42]. Arctiin inhibits growth of the human prostate cancer PC-3 cells, which is associated with an arrest of cyclin D1 expression [43]. However, most of these studies have been conducted in vitro. Our data extends this knowledge by showing that *A. lappa* extract has significant anti-tumoral activity during experimental melanoma. Multiple factors from tumors stimulate the production of inflammatory mediators, such as cytokines [30,44], but also reprogram hematopoiesis towards an expansion in neutrophil output. Activated neutrophils kill tumor cells, but recent evidences demonstrate that neutrophil activation can drive tumor progression and metastasis via stromal remodeling, impairment of T cell-dependent anti-tumor immunity or even by stimulating angiogenesis [45]. Our study demonstrates that besides its direct effect on tumoral cells, *A. lappa* extract suppresses neutrophil migration during LPS or melanoma-induced inflammation. This reduction correlated with lower levels of the inflammatory cytokines IL-6, TNF-α, and IL-1β, reduced edema formation and reduced NO production. Therefore, Alhe might promote resistance to melanoma progression and enhance mice survival by regulating inflammation.

## Figures and Tables

**Figure 1 medicines-06-00081-f001:**
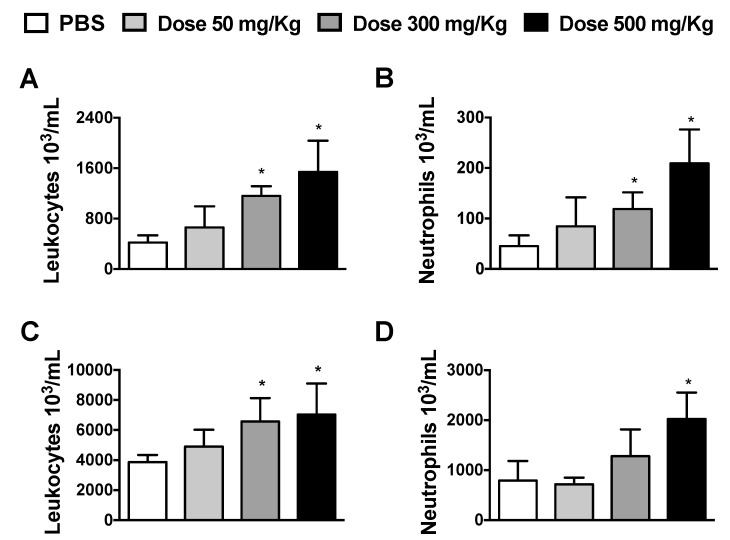
Alhe induces leukocyte accumulation into the peritoneal cavity and peripheral blood. Total leukocytes and neutrophils were evaluated after injection of 300 μL of Alhe at different concentrations (50, 300, and 500 mg/kg) in the peritoneal cavity (**A**,**B**) or peripheral blood (**C**,**D**). The experiment was conducted twice using five mice per group (n = 10); the error bars denote ± SEM. * indicates control PBS versus different doses. These differences were considered significant when *p* < 0.05, according to ANOVA with Tukey’s post-test.

**Figure 2 medicines-06-00081-f002:**
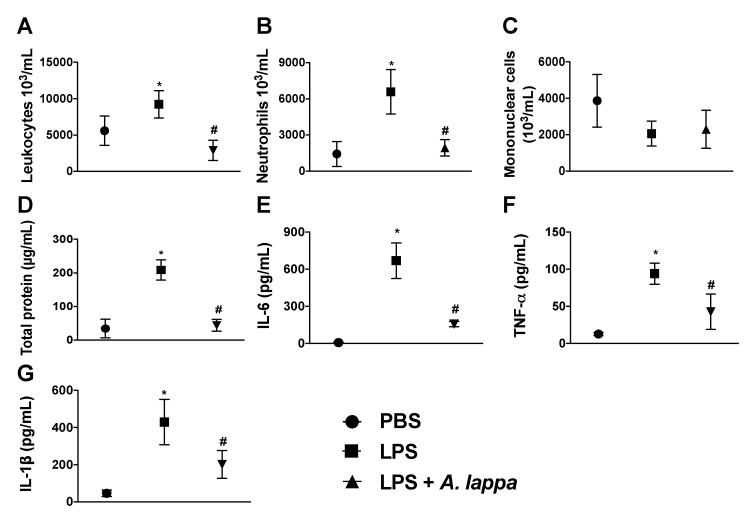
*A. lappa* extract suppresses inflammation in lipopolysaccharide (LPS)-inflamed air pouches. After 1 h of injection of phosphate-buffered saline (PBS) (PBS/1 mL/pouch) or LPS (1 μg/mL/1 mL/pouch) into air pouches, mice were injected with *Alhe* (300 mg/kg/1 mL/pouch) or PBS (PBS/1 mL/pouch). Total and differential leukocyte numbers were determined in the lavage fluid, after 4 h. (**A**) Total leukocytes, (**B**) neutrophils, (**C**) mononuclear cell counts, and (**D**) total protein quantification. (**E**) IL-6, (**F**) TNF-α, and (**G**) IL-1β were quantified by ELISA. The experiment was conducted twice using five mice per group (n = 5), and error bars denote mean ± SEM. * PBS (control) versus LPS; # LPS versus LPS + *A. lappa*. Differences were considered significant if *p* < 0.05 according to ANOVA with Tukey’s post-test.

**Figure 3 medicines-06-00081-f003:**
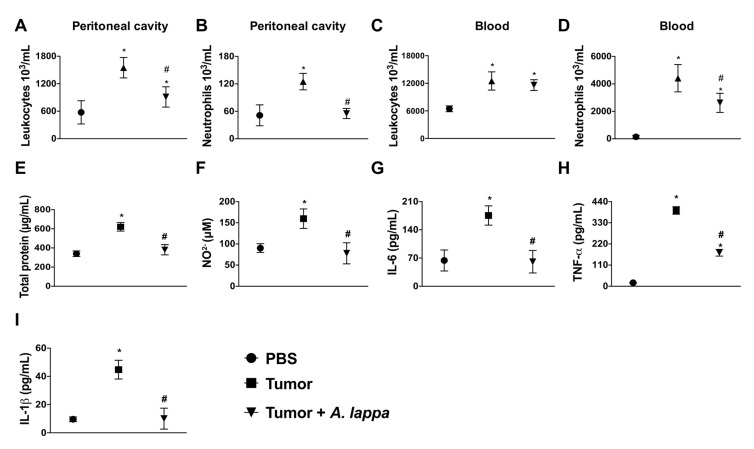
*A. lappa* extract reduced inflammation induced by tumor implantation. (**A**,**C**) Total leukocytes and (**B**,**D**) neutrophil accumulation was evaluated in the peritoneal cavity after 30 days of the B16F10 melanoma implantation. The animals received *Alhe* (50 mg/kg/300 μL) or PBS (300 μL) via i.p. every 5 days for 30 days. (**E**) Total protein, (**F**) NO_2_^−^, (**G**) IL-6; (**H**) TNF-α, and (**I**) IL-1β were quantified in the peritoneal fluid. The experiment was conducted twice using five mice per group (n = 5). Data represents mean ± SEM. * control versus tumor or tumor + *A. lappa*; ^#^ tumor versus *A. lappa*. Differences were considered significant with *p* < 0.05 according to ANOVA with Tukey’s post-test.

**Figure 4 medicines-06-00081-f004:**
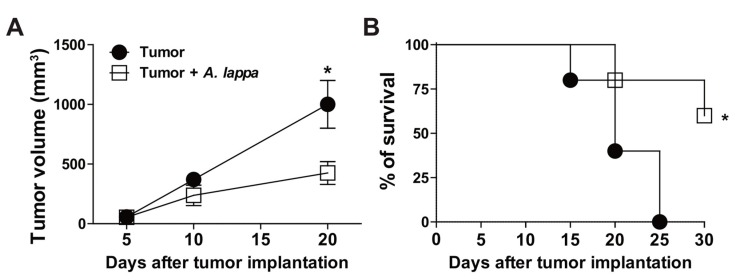
*A. lappa* extract reduces tumor growth and rescues animal mortality. (**A**) For assessment of tumor size and (**B**) mice survival, B16F10 melanoma cells were implanted and the animals received Alhe (50 mg/kg/300 μL) or PBS (300 μL) every 5 days for 30 days. (**A**) Tumors were measured with a Vernier caliper, and tumor volume was calculated with the equation—tumor length × height × width/2. (**B**) Animal survival was monitored for 30 days. The experiment was conducted twice using five mice per group (n = 5). Data denote mean ± SEM. * Tumor versus tumor + *A. lappa*. Differences were considered significant with *p* < 0.05, according to ANOVA with Tukey’s post-test (tumor growth) or the log-rank test (mice survival).

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
