# Peer review of "Arctium lappa Extract Suppresses Inflammation and Inhibits Melanoma Progression"

_medicines, 2019, doi:10.3390/medicines6030081_

Round 1
Reviewer 1 Report
The manuscript entitled “Arctium lappa extract suppresses inflammation and melanoma progression” provides novel knowledge that may contribute to develop the better treatment of acute inflammation and cancer. I read the manuscript with interest, and I think that this study may be had interest by many researchers. However, concerns raised are shown as below.
1. Lignans from A. lappa exhibit antiproliferative and apoptotic effects over leukemia cells, and arctigenin exhibits anti-inflammatory effects. I think that the authors should perform in vivo experiments using these compounds instead of Alhe, too.
2. p.2, lines 72~78
The authors finally obtained 1,000L (997 L PBS + 3 L DMSO) of Arctium lappa extract (concentration: 1 g/mL) from 100 g of ground and dry bark. Really?
3. Injection methods
Alhe was injected ip in this paper. I doubt whether data obtained from ip injection can correctly show the practical effects of Alhe on treatment of inflammation and cancer. I think that ip injection of Alhe cannot use at medical scenes. Therefore, I recommend that the authors carry out oral administration, too.
4. p.3, line 123
NO2- → NO2-
5. Figure 1
I think that these data should be indicated on the bar graph but not line graph.
6. Figure 3D and H
These graphs are cut off in the middle.
7. Legend of Figure 3F
NO → NO2-
8. Figure 4A
The scale of horizontal axis is erroneous. I think that these data also should be indicated on the bar graph but not line graph.
9. p.8, lines 267~269
How about immune activation?
Reviewer 2 Report
Comments
This manuscript better described the inhibitive effects of Arctium lappa extract on the inflammation and melanoma progression in vivo. This manuscript was written very well and understandable. In my opinion, this manuscript can be accepted after minor revision.
Some questions:
1) Abbreviation and full name should be used in its first occurrence, such as LPS (lipopolysaccharides).
2) Page 2, line 76, “997 L PBS” and “3 L DMSO”? Whether “L” is right?
3) In Figure 3, 3D and 3H are partially lost.
4) In the “Introduction”, I suggest the authors add some important references related to the resistance, novel strategies, opportunities, and challenges of cancer chemotherapy, such as (Future Medicinal Chemistry, 2018, 10(16), 1971-1996.; Future Medicinal Chemistry, 2017, 9(4), 403-435.; Cancers, 2019, 11, 317).
Reviewer 3 Report
Manuscript entitled” Arctium lappa extract suppresses inflammation and melanoma progression “investigated the effects of A. lappa hydroalcoholic extract over different models of inflammation in vivo “ which observed significant reduction of neutrophil infiltration and production of inflammatory mediators after LPS or tumoral cell injection in mice that were treated with Alhe. Furthermore, Alhe treatment had a striking impact on melanoma reduction and survival of mice. The results and methods and all experimental work are conducted well but the manuscript needs:
1. extensive language and editing corrections.
2. Why the authors did not separate the active constituents and identified them?
3. More updated references required in the references list.
4. expand your conclusion section.
Reviewer 4 Report
Discussion and hypothesis of manuscript are very weak.The experimental design and the discussion of data are very poor. The set of analyses is scarce. Current presentation of data is still incomplete for a scientific article.
Round 2
Reviewer 1 Report
Although I read the revised manuscript and your comments, some concerns have not solved yet. Remained concerns are shown as below.
1. Injection methods
Even now, I doubt whether data obtained from ip injection can correctly show the practical effects of Alhe on treatment of inflammation and cancer. Because ip injection of Alhe cannot use at medical scenes, the authors should carry out oral administration, too.
2. Figure 4A
The scale of horizontal axis is erroneous (same length: 5 days~10 days, 10 days~20 days).

Round 3
Reviewer 1 Report
My concerns on the manuscript are now responded.